# Modulation of Hoogsteen dynamics on DNA recognition

Yu Xu[1,2], James McSally[3], Ioan Andricioaei[3] & Hashim M. Al-Hashimi [1,2]

In naked duplex DNA, G–C and A–T Watson-Crick base pairs exist in dynamic equilibrium with their Hoogsteen counterparts. Here, we used nuclear magnetic resonance (NMR) relaxation dispersion and molecular dynamics (MD) simulations to examine how Watson-Crick/Hoogsteen dynamics are modulated upon recognition of duplex DNA by the bisintercalator echinomycin and monointercalator actinomycin D. In both cases, DNA recognition results in the quenching of Hoogsteen dynamics at base pairs involved in intermolecular base-specific hydrogen bonds. In the case of echinomycin, the Hoogsteen population increased 10-fold for base pairs flanking the chromophore most likely due to intermolecular stacking interactions, whereas actinomycin D minimally affected Hoogsteen dynamics at other sites. Modulation of Hoogsteen dynamics at binding interfaces may be a general phenomenon with important implications for DNA–ligand and DNA–protein recognition.

[1] Department of Chemistry, Duke University, Durham, NC 27710, USA. [2] Department of Biochemistry, Duke University School of Medicine, Durham, NC 27710, USA. [3] Department of Chemistry, University of California Irvine, Irvine, CA 92697, USA. Correspondence and requests for materials should be addressed to H.M.A.-H. (email: hashim.al.hashimi@duke.edu)

The quest for a molecular understanding of how proteins and ligands recognize DNA has traditionally been pursued based on the assumption that the double helix forms a structure composed entirely of Watson-Crick base pairs (bps). Recent nuclear magnetic resonance (NMR) studies have started to challenge this paradigm by showing that in naked canonical duplex DNA, G–C and A–T Watson-Crick bps exist in a dynamic equilibrium with short-lived (ms lifetimes) low-abundance (populations <1%) Hoogsteen bps[1,2] (Fig. 1). The Hoogsteen bps[1] form by flipping the purine bases 180° around the glycosidic bond followed by constriction of the partner bps by ~2 Å to create a new set of hydrogen bonds (H-bonds) (Fig. 1). This dynamic process, which is sometimes referred to as "Hoogsteen breathing"[3], has been shown to occur robustly in DNA double helices across a wide variety of sequences and positional contexts[2,4–7]. The stability and lifetimes of transient Hoogsteen bps is also strongly dependent on sequence[5] potentially providing new mechanisms for sequence-dependent DNA transactions.

Hoogsteen bps alter the chemical presentation of the bases to external cellular factors[6,8], the electrostatic potential of DNA[9], and its overall shape[10,11]. Consequently, Hoogsteen bps can potentially play unique roles in DNA recognition through both direct and indirect readout mechanisms[12,13]. Indeed, Hoogsteen bps have been observed in a handful of DNA–protein and DNA–small molecule complexes where they appear to play roles in DNA recognition (reviewed in ref[6]). For example, the crystal structure of DNA bound to the TATA box-binding protein (TBP) features two consecutive G–C+ Hoogsteen bps[14], where one of the *syn* guanines has been proposed to alleviate a steric clash with a nearby leucine side chain. Additionally, the crystal structure of DNA in complex with the DNA-binding domain of p53 tumor suppressor protein features two consecutive A–T Hoogsteen bps[9]. The narrowed and more negatively charged minor groove flanking the two A–T Hoogsteen bps has been proposed to form favorable electrostatic interactions with a positively charged arginine residue[9]. Hoogsteen bps have also been observed in DNA–small molecule complexes[15–17], which are proposed to be stabilized by stacking interactions[18,19].

Considering the ease with which Hoogsteen bps can form within the double helix and their preference for stressed regions, particularly sites in which the DNA is underwound and/or kinked toward the major groove[10,11], it comes as a surprise that Hoogsteen bps have not been more broadly observed in structures of DNA–protein and DNA–ligand complexes. Several crystallographic studies have documented difficulties in distinguishing between Watson-Crick and Hoogsteen bps[8,20,21]. Many structures are determined at a resolution that does not permit to detect low-abundance conformations[22]. Some of these ambiguities could arise due to enhanced Hoogsteen breathing. Indeed, prior studies of DNA–ligand complexes indicated that the Hoogsteen bps could be dynamic in nature[17,23].

Changes in the conformational dynamics following recognition of proteins and ribonucleic acids have been shown to play essential roles determining thermodynamic binding affinity[24], kinetics mechanisms of binding[25,26] as well as downstream biological activity[27,28]. Changes in Hoogsteen dynamics on recognition of DNA by proteins or small molecules could potentially be widespread and play important roles determining the binding affinity and specificity, rates of complex formation, as well as downstream biological activity. Here, as a first step toward examining whether or not Hoogsteen breathing is modulated on DNA recognition, we used NMR in concert with molecular dynamics (MD) simulations to characterize changes in Watson-Crick/Hoogsteen breathing that follows recognition by two different small molecules; echinomycin and actinomycin D. Our results expose a new layer of structural plasticity at the interfaces

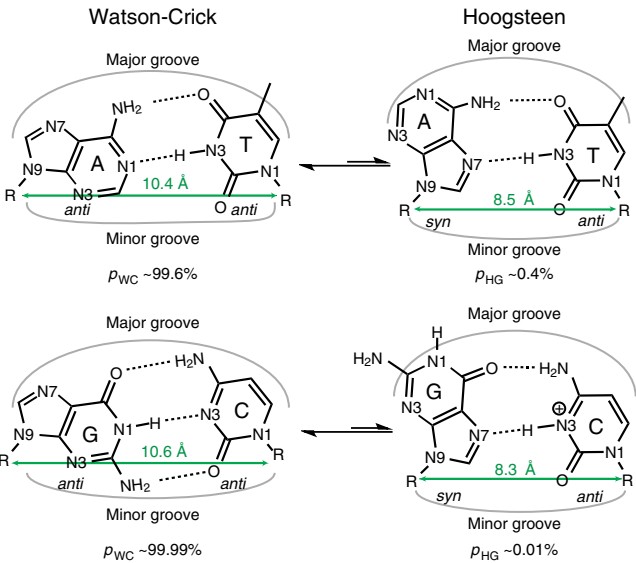

**Fig. 1** Dynamic equilibrium between Watson-Crick and Hoogsteen base pairs. The abundances of both G–C and A–T Hoogsteen at neutral pH were taken from ref.[7]

of DNA complexes that may have general and important implications for DNA recognition.

## Results

**Hoogsteen modulation in the DNA–echinomycin complex.** The peptide antibiotic echinomycin[29–31] was the first molecule shown to bind DNA in a bisintercalative manner and has since served as a paradigm for understanding the principles of DNA bisintercalation by other molecules (Fig. 2a). Echinomycin consists of two quinoxaline-2-carboxylic acid chromophores linked together by a cross-bridged cyclic octapeptide dilactone containing both L- and D-amino acids. It recognizes 5′CpG3′ through base-specific H-bonds between the cyclic peptide linker and the guanine-NH$_2$ and guanine-N3 in the DNA minor groove while the two quinoxaline rings bisintercalate to flank the 5′CpG3′. Prior studies have shown that binding of two echinomycin molecules to CG steps flanking a TA step in sequences such as d(CGTACG) leads to formation of two neighboring A–T Hoogsteen bps within the TA step[16,23] (Fig. 2a). Echinomycin and the related peptide antibiotic triostin A are the only compounds known to induce Hoogsteen bps upon duplex DNA recognition.

We used $^{13}$C and $^{15}$N NMR spin relaxation in the rotating-frame ($R_{1\rho}$) relaxation dispersion (RD)[32–34] to examine how DNA recognition by echinomycin affects Hoogsteen breathing. The RD experiment measures the chemical exchange contribution ($R_{ex}$) to intrinsic transverse relaxation ($R_2$) in the presence of a spinlock field that has variable power ($\omega_{SL}$) and offset frequency ($\Omega$). The data can be used to characterize chemical exchange directed toward low-abundance (populations as low as 0.01%) and short-lived (ms lifetimes) conformational states often referred to as "excited states" (ES)[35].

NMR analysis indicates that the palindromic DNA sequence d-5′-ACA**CG**T**A**CGTGT-3′ (**CG**-binding step and **TA** Hoogsteen bp) forms a complex with echinomycin (Fig. 2b) that is consistent with that reported previously by NMR for a related DNA sequence[23] (Supplementary Note 1). The complex features two echinomycin molecules bound at the two CG steps and tandem T6–A7/A7–T6 Hoogsteen bps flanked by Watson-Crick bps (Fig. 2a and Supplementary Fig. 1). Binding of echinomycin is

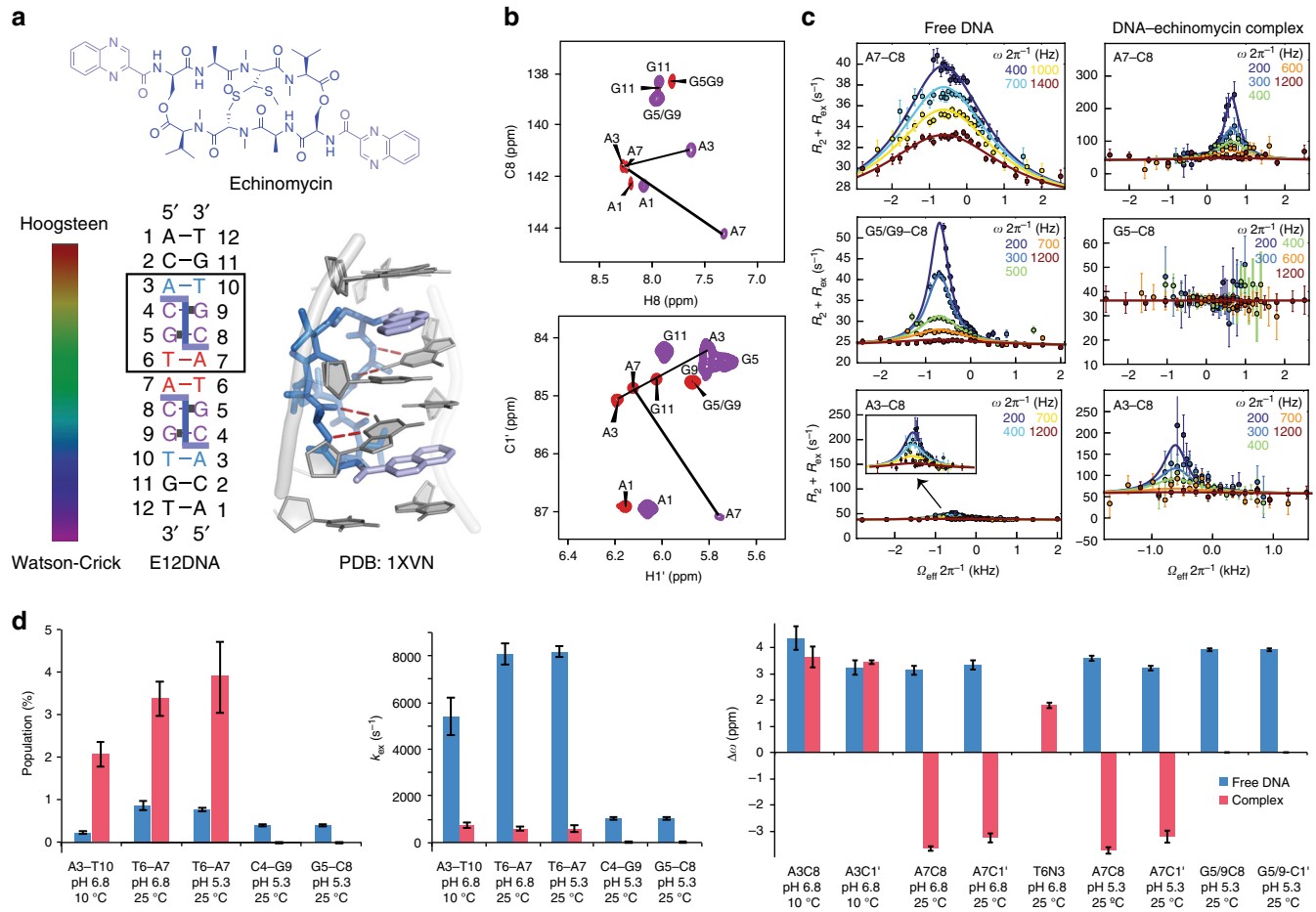

**Fig. 2** Hoogsteen breathing in the DNA–echinomycin complex. **a** Chemical structure of echinomycin, crystal structure of the DNA–echinomycin complex (PDBID: 1XVN), and DNA construct used in NMR studies with color code showing Hoogsteen population (ranging between 0 to 100%). **b** Two-dimensional HSQC NMR spectra of DNA showing chemical shift perturbations on complex formation (red: free DNA; purple: complex). **c** Representative RD profiles measured in the free and bound duplex DNA showing modulation of Hoogsteen breathing on complex formation. Best fits to the Bloch–McConnell equations are shown. Error bars represent experimental uncertainty (one s.d.) derived using a Monte-Carlo method. **d** Comparison of exchange parameters measured in the free DNA and the DNA–echinomycin complex (Supplementary Table 1). RD and exchange parameters for G5/G9 are obtained from measurements on these overlapping resonances arising due to their very similar sequence contexts. Errors in all RD-derived fitted parameters including population, $k_{ex}$ and $\Delta\omega$ reflect experimental uncertainty (one s.d.) calculated by the Monte-Carlo approach from a single RD measurement containing more than 40 data points (see Methods). $\chi^2 < 1.5$, $P < 0.001$

slow on the NMR timescale[36] (Supplementary Fig. 1) consistent with low µM binding affinity[37].

In the free DNA duplex, RD is observed at every non-terminal bp examined consistent with Hoogsteen breathing (Fig. 2c and Supplementary Fig. 2)[2,4–7]. Fitting of the RD data measured at the A3–T10, T6–A7, C4–G9, and G5–C8 bps to a 2-state exchange model yielded chemical shifts (~3 ppm downfield shifted purine-C1′ and purine-C8, and ~2 ppm upfield shifted thymine-N3), populations (0.25–0.85% for AT, ~0.4% for GC) and lifetimes (0.1–0.2 ms for AT, ~1.0 ms for GC at low pH) for transient Hoogsteen bps that are within the range of values reported previously for different DNA duplexes[2,4–7] (Fig. 2d and Supplementary Table 1).

Interestingly, the measured RD profiles were significantly altered in the DNA–echinomycin complex (Fig. 2c and Supplementary Fig. 2). No RD was observed at C4–G9 and G5–C8 bps within the CG-binding site, even when varying the temperature to modulate the rate of exchange and bring any transient state within the RD detection window. This indicates the Hoogsteen population is lower than the detection limit (0.01%). Indeed, in structures of DNA–echinomycin

complexes[16,38,39], the guanine-NH₂ and guanine-N3 of G5 and G9 form base-specific H-bonds with the peptide linker (Fig. 2a). These H-bonds would be disrupted if the guanine base were to flip to form a G(syn)-C⁺ Hoogsteen bp. Therefore, quenching of Hoogsteen breathing at these sites is most likely due to the increased stability of the Watson-Crick relative to the Hoogsteen bp.

In sharp contrast, the RD measured at both A3–C1′ and A3–C8 in the A3–T10 Watson-Crick bp flanking the CG step is significantly enhanced in the complex (Fig. 2c and Supplementary Fig. 2). The Hoogsteen population increases ~9-fold (from 0.24 ± 0.04% to 2.07 ± 0.30% at 10 °C, while the exchange rate decreases ~7-fold (from 5411 ± 807 s⁻¹ to 743 ± 131 s⁻¹) (Fig. 2d). Errors in all RD-derived fitted parameters including population, $k_{ex}$ and $\Delta\omega$ reflect experimental uncertainty (one s.d.) calculated by the Monte-Carlo approach from a single RD measurement containing more than 40 data points (see Methods). Strikingly, the Hoogsteen population reaches 7.9 ± 0.7% at 25 °C. This significant increase in the Hoogsteen population at A3–T10 bp was robustly observed based on both A3–C8 and A3–C1′ RD data across different pH conditions (pH 5.3 and 6.8) and temperatures

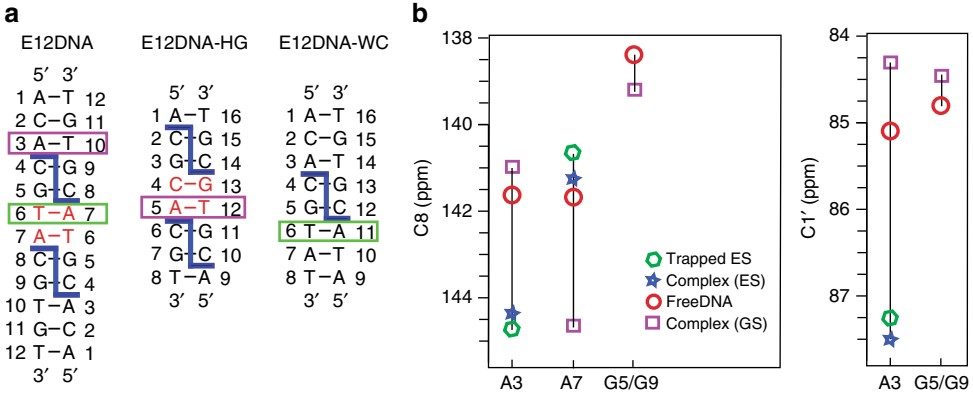

**Fig. 3** Chemical shift fingerprinting alternative transient states. **a** DNA constructs used to trap alternative conformations of bps in the parent E12DNA. The stable Hoogsteen bps are shown in red. **b** Comparison of chemical shifts measured for transient states in the parent complex with those measured for the ground state of the free parent, bound parent, and bound trapped complexes. The RD data for A3C1′ were measured at 10 °C for improved spectral resolution and data quality

(10 and 25 °C) (Supplementary Fig. 2 and Supplementary Table 1). Prior studies showed that when the A3 bp is terminal, it forms a highly stable A–T Hoogsteen bp in echinomycin complexes[17,23,38]. Our data suggest that the dynamic equilibrium in a non-terminal A–T bp is significantly biased toward the Hoogsteen bp as compared to the free DNA. The enhanced Hoogsteen population could be due to favorable stacking interactions between the quinoxaline ring and *syn* adenine base due to the larger dipole moment of the A–T Hoogsteen bp, which have previously been proposed to stabilize both the tandem T6–A7/A7–T6 as well as terminal A–T Hoogsteen bps in DNA–echinomycin complexes[18].

Surprisingly, we also observed significant RD at the tandem T6–A7/A7–T6 Hoogsteen bps in the DNA–echinomycin complex (Fig. 2c and Supplementary Fig. 2). Fitting of the RD data at 25 °C yielded a transient state with population of $3.37 \pm 0.41\%$, exchange rate ($k_{ex} = k_1 + k_{-1}$) of $597 \pm 88\ \mathrm{s}^{-1}$ (compared to population of $0.87 \pm 0.11\%$ and $k_{ex}$ $8089 \pm 462\ \mathrm{s}^{-1}$ in the free DNA) and oppositely shifted chemical shifts that are directed toward the chemical shifts of canonical Watson-Crick bps (Fig. 2d). This slower exchange process (note that RD for this process is not observed at a lower temperature of 10 °C most likely because the exchange is too slow and falls outside detection) is consistent with inversion of the equilibrium and reverse chemical exchange between Hoogsteen and Watson-Crick in the DNA–echinomycin complex. To our knowledge, this is the first definitive observation of exchange between a major Hoogsteen and minor Watson-Crick conformation. However, it should be noted that prior NMR studies by Feigon et al.[23,39] showed these Hoogsteen bps to be highly dynamic and this flexibility was proposed to originate from either base opening or Hoogsteen to Watson-Crick transitions.

The exchange parameters for Hoogsteen breathing vary significantly across the different bps in the DNA–echinomycin complex (Fig. 2d). This heterogeneity is inconsistent with a global process such as dissociation of echinomycin, which would be expected to give rise to similar exchange parameters at different sites. We were also able to rule out dissociation of echinomycin as a major source of RD based on concentration-dependent RD measurements (Supplementary Fig. 2 and Supplementary Note 2) as well as analysis of chemical shifts (see below).

Finally, we note that in the free DNA, Hoogsteen breathing could only be observed at the two terminal bps (A1–T12 and C2–G11) when lowering the temperature to 10 °C (and pH to 5.3 in the case of G–C+ Hoogsteen) so as to slow down exchange to within the RD detection limits (Supplementary Fig. 2).

Interestingly, in the DNA–echinomycin complex, an alternative transient state that is inconsistent with Hoogsteen is observed at these terminal bps under neutral pH of 6.8 (Supplementary Fig. 2). This exchange process does not impact Hoogsteen breathing at other sites and likely reflects non-specific binding of echinomycin to the terminal AC step in the DNA (Supplementary Note 2).

**Trapping the echinomycin bound DNA transient states**. To confirm the identity of the proposed transient alternative bps in the DNA–echinomycin complex, we prepared DNA constructs that are designed to predominately (population >90%) form the alternative bp conformations when bound to echinomycin. We then compared the resultant DNA chemical shifts in these "trapped" complexes with those measured for the transient state using RD measurements on the parent complex. These studies also provide an independent means of examining the feasibility of forming echinomycin complexes with the proposed alternative bp conformations.

The NMR RD data indicate that in the complex, the tandem T6–A7/A7–T6 Hoogsteen bps transiently form Watson-Crick bps. It has been shown that formation of these tandem Hoogsteen bps requires that a TA step be sandwiched by two bound echinomycin molecules[18,23,38,40]. We generated complexes with the TA step in a Watson-Crick conformation by omitting one of the two CG-binding sites. The non-palindromic sequence (E12DNA-WC) containing a single CG-binding site (5′-ACAC**CG**-**TA**T-3′, Fig. 3a) experiences significant DNA chemical shift perturbations in and around the binding site that are similar to those observed for the corresponding residues in the parent DNA–echinomycin complex (Supplementary Fig. 3). However, in contrast to the parent complex, 2D heteronuclear single quantum coherence (HSQC) NMR spectra clearly show that T6–A11 is predominantly Watson-Crick at room temperature (Supplementary Fig. 3). The resultant bound chemical shift for A11C8 in the T6–A11 bp is in good agreement with those of A7C8 measured for the transient state by RD in the parent complex (Fig. 3b). These chemical shifts are also distinct from those of the free DNA (Fig. 3b) helping rule out partial dissociation of echinomycin as a source of the observed RD.

We used 5′-A**CGCA**C**G**T-3′ (E12DNA-HG, Fig. 3a) to mimic the transient A3–T10 Hoogsteen bp in the parent complex. Unlike for the transient state in the parent complex, the trapped A–T Hoogsteen bps is neighbored by a Hoogsteen rather than Watson-Crick C–G bp (Fig. 3a). At pH 5.3, this complex forms

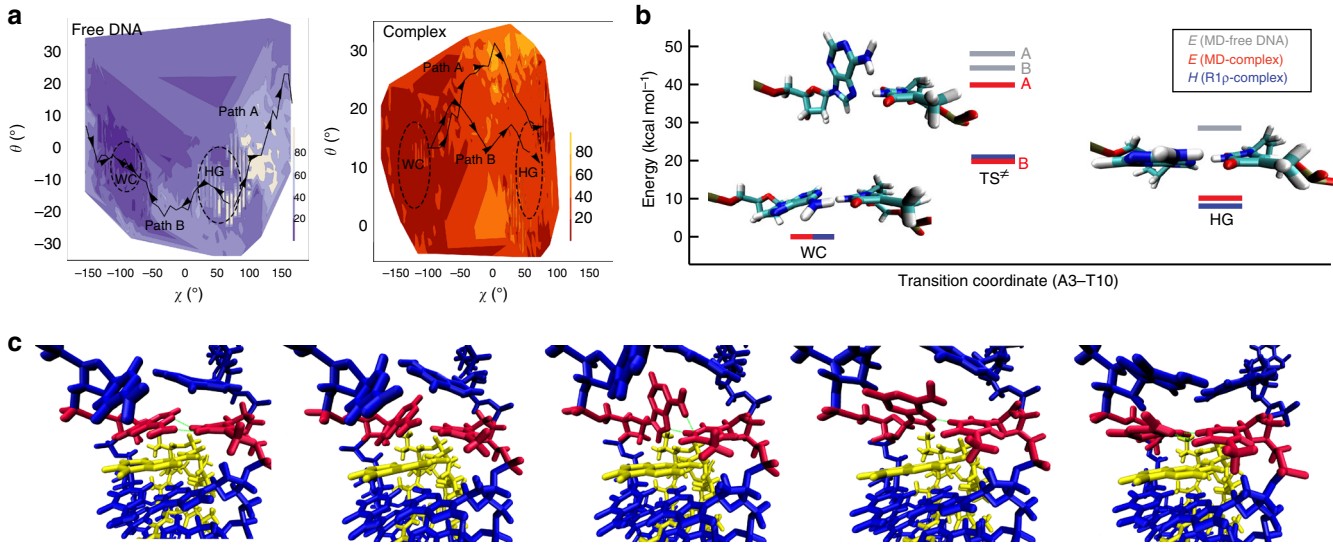

**Fig. 4** Biased MD simulations at A3–T10 in the DNA–echinomycin complex. **a** Contour plots showing the relative interaction energy ($E$ kcal mol$^{-1}$) as a function of the base opening ($\theta$) and flipping ($\chi$) angles from multiple trajectories. **b** Initial Watson-Crick, final Hoogsteen, and transition state (path B) structures for A3–T10 illustrating the span of the transition barriers and their relative interaction energies in free DNA (gray) and in the DNA–echinomycin complex (red), respectively. Enthalpies ($H$) derived from temperature-dependent NMR RD data are shown in blue for comparison with experiment. **c** Snapshots from a single representative transition pathway (path B) for A3–T10 in the DNA–echinomycin complex

tandem C$^+$4–G13/ A5–T12 Hoogsteen bps[41] as verified by 2D nuclear Overhauser effect spectroscopy (NOESY) and 2D HSQC spectra (Supplementary Fig. 3). The A5C8 and A5C1′ chemical shifts measured in this trapped complex are in good agreement with the corresponding A3C8 and A3C1′ chemical shifts measured for the transient state by RD in the parent complex (Fig. 3b).

These results support our assignment of the transient states as well as the feasibility of forming echinomycin complexes with these alternative DNA bp configurations.

**Biased and unbiased molecular dynamics simulations.** We used both biased and equilibrium MD simulations[2,4] to further examine whether or not Hoogsteen to Watson-Crick and Watson-Crick to Hoogsteen transitions are indeed stereochemically feasible in the presence of a nearby bound echinomycin molecule. Simulations were not carried out for C4–G9 and G5–C8 bps, because formation of G–C$^+$ Hoogsteen bps is coupled to cytosine protonation[7] through a process that could involve multiple proton transfer events that are difficult to model computationally in complex environments. Rather, all biased MD simulations of the Watson-Crick to Hoogsteen transition have focused on A–T not G–C[42,43]. Note that while separate simulations could be conducted for the individual Hoogsteen or Watson-Crick bps, these would not provide information regarding the ease of their inter-conversion, which is the key information we are interested in obtaining from these computational studies.

Thirty biased simulations were run, each starting with a different initial velocity. Control simulations on A3–T10 in the free DNA duplex resulted in several (4 of 30 simulations) successful transitions between Watson-Crick and Hoogsteen resulting in a Hoogsteen conformational landscape similar to those reported previously using a similar method[4] (Fig. 4a and Supplementary Movie 1). By comparison, all 30 simulations resulted in successful transitions between Watson-Crick and Hoogsteen without echinomycin dissociation in the DNA–echinomycin complex. Relative to the free DNA, the simulations indicate a lower barrier height for the Watson-Crick

to Hoogsteen transition in the complex DNA as well as greater enthalpic stabilization of the Hoogsteen bp (Fig. 4b). Indeed, both the barrier heights and energetic differences computed by MD are in very good agreement with the NMR RD measured counterparts (Fig. 4b).

Among two pathways (pathways A and B) sampled by the trajectories, the more favored (pathway B) is similar to that observed in the free DNA[2], with the exception that rotation of the base about the glycosidic bond is predominantly clockwise in the complex, whereas a mixture of clockwise and counter-clockwise rotations are observed in the free DNA (Fig. 4a). The transition state features a purine base that is near-orthogonal to its paired pyrimidine resulting in disruption of the Watson-Crick bp as well as stacking interactions with the quinoxaline rings, which remains stacked on the flanking G–C bps (Fig. 4c). The base flipping appears to be unhindered due to the flexibility of the neighboring base pairs that exhibit collective conformational changes during the transition (Supplementary Movie 2).

Although modeling the reverse transition from Hoogsteen to Watson-Crick in the tandem T6–A7/A7–T6 bp is complicated by evidence showing that these bps may form cooperatively[18,41,44], leading potentially to complex transition pathways where two bps flip simultaneously, simulations in which a single bp was flipped did successfully transition from Hoogsteen to Watson-Crick without disrupting the bound echinomycin (Supplementary Fig. 4 and Supplementary Movies 3 and 4). This supports the feasibility of having Hoogsteen to Watson-Crick transition in the presence of the bound echinomycin.

**Hoogsteen modulation in the DNA–actinomycin D complex.** To examine whether modulation of Hoogsteen breathing is a general phenomenon in DNA–small molecule recognition, we studied a complex of DNA bound to the actinomycin D[45–49] (Fig. 5a). Actinomycin D shares many features with echinomycin. The two peptide antibiotics have a similar structure (Fig. 2a and Fig. 5a). Both bind to DNA in the minor groove containing at least two G–C bps, forming base-specific contacts with the guanine-NH2 and guanine-N3, and with actinomycin D recognizing 5′GpC3′ as compared to 5′CpG3′ in the case of

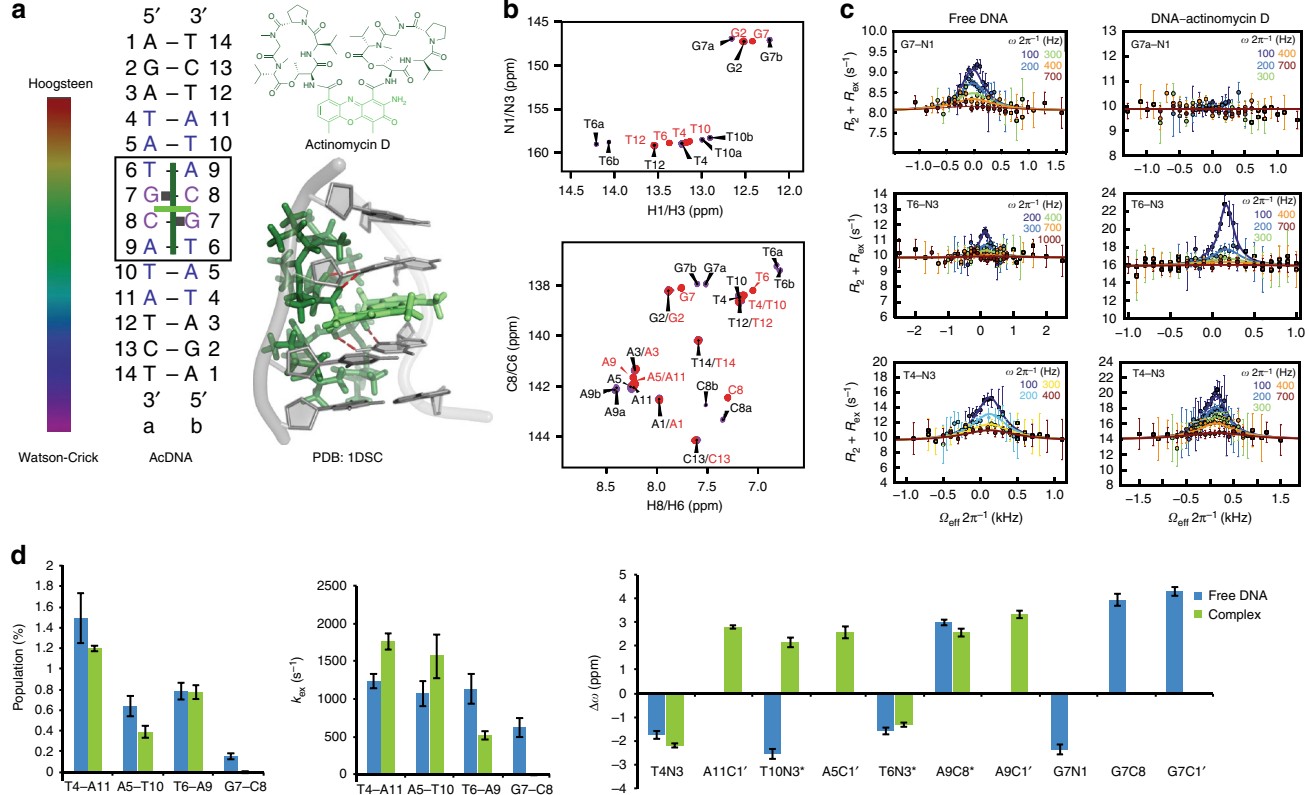

**Fig. 5** Hoogsteen breathing in the DNA–actinomycin D complex. **a** Chemical structure of actinomycin D, crystal structure of the DNA–actinomycin D complex (PDBID 1DSC), and DNA construct used in NMR studies with color code showing Hoogsteen population (ranging between 0 to 100%). **b** Two-dimensional HSQC NMR spectra of DNA showing chemical shift perturbations on complex formation (red: free DNA; purple: complex). **c** Representative RD profiles measured in the free and bound duplex DNA showing modulation of Hoogsteen breathing on complex formation. Best fits to the Bloch–McConnell equations are shown. Error bars represent experimental uncertainty (one s.d.) derived using a Monte-Carlo method. **d** Comparison of exchange parameters measured in the free DNA and the DNA–actinomycin D complex. Sites with two sets of degenerate resonances are indicated with a "*". RD data are shown for the "a" set of resonances noting that very similar results were obtained for the corresponding "b" set of resonances (Supplementary Table 1). Errors in all RD-derived fitted parameters including population, $k_{ex}$ and $\Delta\omega$ reflect experimental uncertainty (one s.d.) calculated by the Monte-Carlo approach from a single RD measurement containing more than 40 data points (see Methods). $\chi^2 < 1.5$, $P < 0.001$

echinomycin. However, in stark contrast to echinomycin, actinomycin D is a monointercalator, which inserts a single phenoxazone chromophore between two G–C bps. Therefore, comparison of echinomycin and actinomycin D may provide insights into how Hoogsteen breathing is differentially modulated by two general classes of DNA binders.

We studied the 1:1 complex between the DNA sequence 5′-AGATATGCATATCT-3′ and actinomycin D. This palindromic sequence contains one favorable binding site, but is also long enough to study effects on Hoogsteen breathing in neighboring bps. NMR analysis of this complex shows that it forms the previously described structure of a similar sequence[50] with all bps being Watson-Crick and with one actinomycin D bound to the G7C8 step (Fig. 5b, Supplementary Fig. 5 and Supplementary Note 3). The NOESY H8/H6-C1′ base–sugar connectivity and imino-imino connectivity in the complex are interrupted at the G7C8 step, consistent with the insertion of the phenoxazone chromophore between the GC step. In addition, two distinct sets of resonances (labeled "a" and "b" in Fig. 5b and Supplementary Fig. 5) are observed in and around the otherwise symmetrical DNA-binding site, which reflect two distinct conformational species that differ with respect to the orientation of the intercalating asymmetric phenoxazone chromophore[47,51]. RD data were measured for both sets of resonances and the fitted exchange parameters are summarized in Supplementary Table 1.

In the free DNA, RD was observed at every bp examined (T4–A11, A5–T10, T6–A9, and G7–C8 bps) consistent with Hoogsteen breathing (Fig. 5c, Supplementary Fig. 2, and Supplementary Table 1). Similar to echinomycin, binding of actinomycin D quenched RD at G7–C8 (Fig. 5c and Supplementary Fig. 2), which forms base-specific H-bonds with actinomycin D (Fig. 5a). These H-bonds would also likely be disrupted with a *syn* guanine base in a Hoogsteen bp. Interestingly, RD was observed at the sugar G7–C1′, but the exchange parameters are inconsistent with Hoogsteen. In particular, two opposite shifts (± 1.5 ppm) are observed for the pseudo-symmetric G7–C1′ chemical shifts (Fig. 5d). The nature of this process requires further investigation but it could involve transitions between the two pseudo-symmetric DNA states.

In contrast to echinomycin, binding of actinomycin D had little effect on the exchange parameters measured at the flanking T6–A9 bp (Fig. 5d, Supplementary Fig. 2 and Supplementary Table 1). In the case of the bisintercalating echinomycin, the two chromophores stack with flanking residues (A3–T10 and T6–A7) that are not directly stabilized by base-specific contacts. In contrast, in the case of actinomycin D, the single chromophore does not stack with the T6–A9 bp, but only stacks between two G–C bps, which are stabilized by base-specific contacts that favor the Watson-Crick geometry. This provides additional support for the importance of stacking interactions with the *syn* purines in promoting Hoogsteen breathing in the DNA–echinomycin complex. However, we cannot

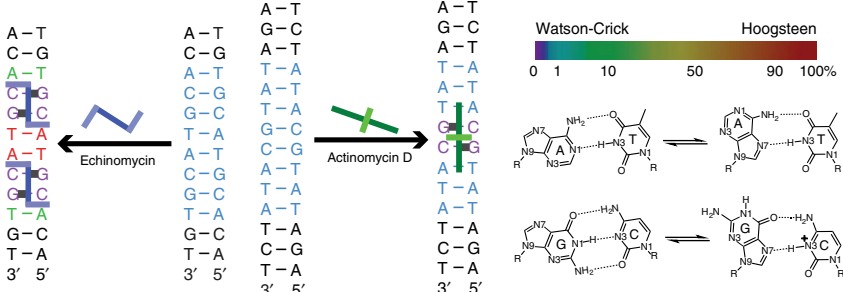

**Fig. 6** Modulation of Hoogsteen breathing upon DNA recognition by ligands. Base-specific H-bonds with the peptide are shown using a thick line. The color code shows Hoogsteen population of individual bp (ranging between 0 and 100%)

rule out that other factors such as differences in degree of unwinding upon complex formation or DNA sequence context can also contribute to these different behaviors.

Interestingly, high-resolution structures show Van der Waals (VDW) contacts between the A9 base/T6 sugar and the peptide backbone of actinomycin D, which could potentially be lost when flipping A9 to a *syn* conformation when forming a Hoogsteen bp. However, our data revealed that the Hoogsteen population at the two pseudo-symmetric T6–A9 bps in the DNA–actinomycin D complex (0.60 ± 0.09%, 0.77 ± 0.07%) is comparable to that in the free DNA (0.79 ± 0.08%) (Fig. 5d). This indicates that the VDW contacts are either weak or that there are other compensatory contacts that can form with the Hoogsteen bp. In addition, the T4–A11 and A5–T10 bps, which are remote from the actinomycin D-binding site, show insignificant changes in Hoogsteen breathing relative to free DNA (Fig. 5d and Supplementary Fig. 2). Interestingly, the T10N3 chemical shift of the Hoogsteen bp is unusually upfield shifted (Fig. 5d) potentially reflecting a unique conformation in the complex. Therefore, DNA recognition by actinomycin D leads to the more surgical arrest of Hoogsteen breathing at sites of direct readout.

The observed impact of actinomycin D binding on Hoogsteen breathing could not be examined computationally because of the difficulties in modeling formation of G–C$^+$ Hoogsteen and because crystal structures are not available for the specific DNA sequence used here.

## Discussion
Our results show that DNA recognition by small molecules can lead to complex modulation of Hoogsteen breathing (Fig. 6). Both echinomycin and actinomycin D quench Hoogsteen breathing at sites involved in direct readout of purine bases. Here, flipping the purine base to a *syn* conformation would not leave appropriate groups in the minor groove to act as either H-bond donors or acceptors. It is very likely that Hoogsteen breathing will also be quenched through direct readout mechanisms in protein–DNA complexes, with potentially important implications for binding affinity and specificity, particularly in cases where the Hoogsteen breathing is significant or where multiple bps are affected simultaneously.

In contrast, only echinomycin promoted Hoogsteen breathing at flanking bps most likely because the two chromophores in the bisintercalator can stack with residues not directly stabilized by base-specific H-bonds (Fig. 6). It is plausible that other intercalators will promote Hoogsteen breathing by similar mechanisms. Such enhancement of Hoogsteen could also be important thermodynamic determinant of DNA-binding affinities. Further studies are required to more broadly assess how the modulation

of Hoogsteen breathing affects the DNA–ligand binding affinities and specificities.

The modulation of Hoogsteen breathing could also be an important determinant of the activity of DNA-binding ligands. Indeed, prior studies showed that echinomycin binds to a sequence element 5′-(T/A)ACGTG-3′ similar to that studied here, which is found in the VEGF hypoxia response element (HRE) in HIF-1 activation[52]. Interestingly, it was shown that echinomycin inhibits VEGF and FLT1 expression to a much greater degree than a designed hairpin polyamide, which binds the same sequence with ~100-fold higher binding affinity[52]. The greater potency of echinomycin was proposed to arise from its more substantial effects on DNA structure, including unwinding and lengthening of DNA helix. The enhancement of Hoogsteen breathing observed here for a similar sequence could be an additional explanation for the greater potency of echinomycin relative to the polyamide.

Our findings emphasize the importance of stacking interactions in stabilizing A–T Hoogsteen bps in the DNA–echinomycin complex. Interestingly, stacking interactions have not been invoked to explain the observation of Hoogsteen bps in crystal structures of protein–DNA complexes. We, therefore, re-examined these six crystal structures and found evidence for stacking in two. In the DNA–TBP complex[14], a phenylalanine (F57) inserts between two G–C$^+$ Hoogsteen bps, disrupting their stacking interactions, and partially stacks with the *syn* guanine (Supplementary Fig. 6). In the complex between dmc very-short-patch repair (Vsr) DNA endonuclease and a cleaved authentic hemi-deaminated/hemi-methylated *dcm* sequence, two trypto-phan residues (W68 and W86) partially stack with the Hoogsteen A–T bp[53] (Supplementary Fig. 6). While a Hoogsteen bp is observed in an equivalent free DNA site, this bp may be stabilized by electrostatic interactions with an arginine residue[9]. Indeed, no A–T Hoogsteen bps are observed in structures of a similar complex that does not contain methylated cytosine, and correspondingly, the A–T bp in these structures do not feature the stacking interactions with the two tryptophan residues. Rather, a phenylalanine side chain (F67) inserts and stacks over the A–T Watson-Crick bp[54] (Supplementary Fig. 6). Considering the many mechanisms available to stabilize Hoogsteen bps (stacking, steric effect, and electrostatics), and that they favor distorted regions of DNA structure, it seems highly likely that DNA–protein recognition can lead to the modulation of Hoogsteen breathing.

Our study focused on DNA–small molecule complexes, in part because of technical difficulties in applying NMR RD experiments to larger protein–DNA complexes but also because the simplicity of the systems afforded us a unique opportunity to verify that the observed chemical exchange does indeed correspond to

Hoogsteen breathing within the context of the complex. Future studies should build on advances in RD methods to study larger protein systems[55] as well as employ other methods such as infrared spectroscopy[41]. This will provide insights into the importance of Hoogsteen modulation on DNA recognition and function.

In conclusion, our results uncover a new layer of complexity in DNA recognition that involves the modulation of the fundamental Hoogsteen breathing motional modes. The design of small molecules that modulate Hoogsteen breathing may open the door for new approaches to target DNA in the development of anti-cancer therapeutics.

## Methods

**Preparation of NMR samples.** Uniformly $^{13}$C/$^{15}$N labeled DNA samples were prepared through the Zimmer and Crothers primer-extension approach[56] using uniformly $^{13}$C-$^{15}$N-labeled deoxyribonucleotide triphosphates (100 mM, Silantes), Klenow fragment (5k U mL$^{-1}$, Thermo Fisher Scientific), and synthetic DNA templates (1 mM, Integrated DNA Technologies). The template contains a ribonucleoside (rU), which is used to release the synthetic oligo product from the template during the product purification. The sequence of the template for E12DNA and AcDNA is ACACGTACGTGT-*AGATCCGAAAGGATCrU* and AGATATGCATATCT-*AGATCCGAAAGGATCrU*, respectively, (the conserved part of the sequence is shown in Italic). Reaction was incubated at 37 °C overnight in 3 mL, 0.5 M Tris buffer (pH 8.8, 1 M NaCl, 0.25 M MgCl$_2$, 0.25 M DDT) and then stopped by heating at 75 °C for 30 min. NaOH (0.3 M in final solution) was added to the reaction mixture, which was then heated at 55 °C for 3 h to release the target oligo product from the template. Reaction mixture was filtered to remove excess pyrophosphate and concentrated down to 1 mL in a 3 kDa cutoff centrifugal concentrator (EMD Millipore). Sample was mixed with equal volume of a formamide-based denaturing DNA loading dye, denatured at 95 °C for 10 min and loaded onto a denaturing polyacrylamide gel electrophoresis gel (20% polyacrylamide/8 M urea), and run overnight to resolve target oligonucleotide from template and other side products. Target band was excised under a ultraviolet hand-lamp followed by electroelution into 20 mM Tris buffer, pH 8, and ethanol precipitation. Sample purity was confirmed using gel electrophoresis (20% polyacrylamide/8 M urea) stained with SybrGOLD before buffer exchange[57]. All nucleotide types were labeled in the case of AcDNA, whereas only adenine and guanine were labeled in the case of E12DNA to aid spectral resolution. All other DNA samples were unlabeled and purchased from IDT with standard desalting. The DNA was dissolved in NMR buffer, denatured at 95 °C for 10 min, and annealed at room temperature. The DNA samples were then buffer-exchanged three times with a centrifugal concentrator (EMD Millipore) to ensure that the final samples contained >99.9% of the desired buffer, which consisted of 15 mM sodium phosphate, 125 mM NaCl, 0.1 mM EDTA, pH 5.3 or 6.8, and 8% D$_2$O unless stated otherwise. Echinomycin and actinomycin D were purchased from Sigma-Aldrich. The DNA–echinomycin sample was prepared by mixing echinomycin dissolved in methanol with the DNA in NMR buffer, followed by slow evaporation of solvent using a stream of air[23]. The dried samples were then redissolved into H$_2$O and 8% D$_2$O. The DNA–actinomycin D sample was prepared by directly adding the solid actinomycin D to DNA sample in NMR buffer.

**NMR experiments.** NMR data were collected on a 600 MHz Varian Inova NMR spectrometer equipped with a Bruker HCPN cryogenic probe; a 700 MHz Bruker Avance III spectrometer equipped with a triple-resonance HCN cryogenic probe; and an 800 MHz Varian DirectDrive2 spectrometer equipped with a triple-resonance HCN cryogenic probe. Data were processed and analyzed with NMRpipe[58] and SPARKY[59], respectively. Resonances were assigned with conventional 2D heteronuclear multiple quantum coherence (HMQC), HSQC, and NOESY experiments.

$^{13}$C and $^{15}$N $R_{1\rho}$ RD experiments[34] were performed at 700 MHz (16.4 T) on Bruker spectrometers with spinlock powers ($\omega_{SL}$, $2\pi^{-1}$ Hz) and offset frequencies ($\Omega$, $2\pi^{-1}$ Hz) listed in Supplementary Data 1. Magnetization of the spins of interest was allowed to relax under an applied spinlock for the following durations: 0–120 ms for N1/N3; 0–60 ms for C8/C1′.

**Analysis of $R_{1\rho}$ data.** Experimental $R_{1\rho}$ relaxation rate constants were calculated by fitting peak intensities vs. relaxation delay durations to a single exponential decay[60]. Uncertainty in the fitted $R_{1\rho}$ values (one s.d.) were derived using the Monte-Carlo method. $R_{1\rho}$ data were fitted to simulated $R_{1\rho}$ values given by the solution to the 2-state (no minor exchange) Bloch–McConnell (BM) equations[61] at each given $\omega_{SL}$ and $\Omega$ value. Residual sums of squares were minimized with a bounded least-squares algorithm[62] yielding best-fit exchange parameters (center value) (Supplementary Table 1). The uncertainty in the exchange parameters was calculated using the Monte-Carlo approach[34]. Basically, 1000 child data sets were generated using the BM equation, the fitted exchange parameters from the parent data set, and the $\omega_{SL}$ and $\Omega$ values of the parent data set. Each child data set was

noise corrupted according to the $R_{1\rho}$ error from the parent data set. The child data sets were then fitted to the BM equation to obtain their individual exchange parameters. Finally, the uncertainty in the exchange parameters was determined by calculating the standard deviation of the individual parameters fitted from the child data sets relative to parameters fitted from parent data set. A two-state exchange model was used to fit the $R_{1\rho}$ RD profiles with the initial magnetization aligned along the effective field of either the ground or average state[62]. The Bayesian information criterion and Akaike information criterion[63] were used in model selection. For most RD data, both protocols yielded acceptable fits and similar exchange parameters (within error). However, for A7C8, A7C1′, A3C8, A3C1′ in DNA–echinomycin complex, both protocols yielded acceptable fits but resulted in different exchange parameters, given the slower exchange rate $k_{ex}$ ~600 s$^{-1}$. The exchange parameters obtained from fits to average state alignment were selected based on a 2-state van't Hoff analysis[4] (Supplementary Fig. 2). In the case of T6N3 in the free and DNA–actinomycin D complex, results from the ground state alignment were reported due to the smaller error in fitted parameters though results from the two fits were within error.

**Biased and unbiased molecular dynamics simulations.** Coordinates for the E12DNA–echinomycin complex were obtained by downloading the 1XVN structure[38] from the Protein Data Bank (PDB). The coordinates of the DNA portion of the complex were loaded into the CHARMM molecular modeling package and coordinates for the terminal two bps were generated using internal coordinate tables within CHARMM[64]. Both A3 bases were rotated 180° at the glycosidic bond to begin in the Watson-Crick conformation. Structures for control simulations of free DNA were generated through the use of make-na[65]. Each A7 was rotated 180° along the glycosidic bond to begin in the Hoogsteen conformation.

The coordinates of a single echinomycin molecule were loaded into Schrodinger's Maestro program[66], to generate bond parameters. Bond parameter and coordinate information for echinomycin were entered into CHARMM CgenFF's automated atom typing program for generation of CHARMM force field parameters for the echinomycin[67–70]. The DNA–echinomycin complex and control free DNA were each placed into cubic water boxes with side lengths of 87 Å with 20,440 and 20,558 TIP3 water molecules, respectively. To insure the neutrality of each system 31 Na$^+$ cations and 9 Cl$^-$ anions were added as well.

Each system was equilibrated using constant temperature and pressure dynamics. Temperature was maintained at 300 K and pressure at 1 atm using the Nose-Hoover Thermostat[71]. Particle-mesh Ewald summation[72,73] was used with cutoffs of 14 Å to calculate electrostatic potentials. Equilibration for each system ran for 300 ps using a leap verlet algorithm. From the final structure produced from the equilibration, for each system, 30 simulations were run under the exact same conditions of equilibration while varying the initial starting velocities sampling the immediate space near either the Watson-Crick or Hoogsteen states.

The biased MD method[74] implemented in the CHARMM package was used to assess conformational transitions between Watson-Crick and Hoogsteen bps for A3 and A7 both in the presence and absence of echinomycin, using a biasing potential $W(\rho(t))$ applied according to equation (1),

$$W(\rho(t)) = \begin{cases} \frac{\alpha}{2}\left(\rho(t) - \rho_a(t)\right)^2, & \text{if } \rho(t) < \rho_a(t) \\ 0, & \text{if } \rho(t) \geq \rho_a(t) \end{cases} \quad (1)$$

where

$$\rho(t) = \left(\frac{1}{N(N-1)}\right) \sum_{i=1}^{N} \sum_{j\neq 1}^{N} \left(r_{ij}(t) - r_{ij}^R\right)^2 \quad (2)$$

and

$$\rho_a(t) = \max_{0 < \tau < \infty} \rho(t) \quad (3)$$

where $\rho(t)$ is a collective distance between a reaction coordinate ($r_{ij}$) and a reference structure ($r_{ij}^R$), and $\alpha$ the force constant of the half-harmonic bias in kcal mol$^{-1}$ Å$^{-4}$. In all cases, biases were placed between pairs of atoms that share a hydrogen-bond in the target structure, ensuring that the adenine base would flip 180° in the $\chi$-direction, and form the appropriate hydrogen-bonding structure of the target conformation.

Trajectories were post-processed in CHARMM, outputting the $\chi$ and $\theta$ angle dependence of the relative interaction energy value (this excludes the bias potential for the biased simulations). The relative interaction energy was calculated for the bp that includes the flipping base as well as the bps above and below the flipping base. In this calculation, each atom of the base was evaluated individually for both bonded and non-bonded terms for the CHARMM force field, which includes the interaction with the ligand and the solvent effect. The ($\chi$, $\theta$, energy) points were binned into a 50 × 50 grid of bins for both angles and the mean of the energy was evaluated within each bin. Contour plots of relative interaction energy as a function of both $\theta$ and $\chi$ were generated.

**Data availability**. All relevant data and the Python code for fitting the $R_{1\rho}$ data are available from the authors (ha57@duke.edu).

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

## Acknowledgements

We thank Dr. Issac Kimsey and Dr. Huiqing Zhou as well as other members of the Al-Hashimi laboratory for their critical input and comments on the manuscript. This work was supported by the US National Institutes of Health grant (R01GM089846). The research reported in this article was performed by the Duke University faculty, research associate and was funded by US National Institute of Health contract to H.M.A.-H.

## Author contributions

Y.X. and H.M.A.-H. conceived the research and experimental design and performed data interpretation. Y.X. carried out all of the experimental work with assistance from H.M.A.-H.; J.M. performed all of the computational work with assistance from I.A. Y.X., J.M., I.A. and H.M.A.-H. wrote the manuscript.

## Additional information

**Competing interests:** H.M.A.-H. is an advisor to and holds a financial interest in Nymirum, an RNA-based drug discovery company. The remaining authors declare no competing interests.

