## [Peer Review File · Nature Communications]

Reviewers' Comments:

Reviewer #1:

Remarks to the Author:

Xu and colleagues use NMR methods and MD simulations to study the phenomenon of "Hoogsteen breathing" in DNA which is characterized by a dynamic equilibrium between standard Watson-Crick (WC) base pairs and non-standard Hoogsteen base pairs. Their study is based on extensive prior work by the Al-Hashimi laboratory and others (in part going back to the 1980s) on the DNA duplex binding of the intercalating drugs echinomycin, actinomycin D and triostin A. In this prior work it was observed that ligand binding is associated with a stabilization of WC C·G base pairs enclosed by the bis-intercalator echinomycin and the formation of adjacent Hoogsteen base pairs.

After the basic structural features of echinomycin and actinomycin A DNA binding are established, the emphasis of the present study is on characterizing the WC/Hoogsteen dynamics and its modulation by intercalator binding. WC/Hoogsteen populations are measured by NMR relaxation dispersion (RD), a method that can detect short-lived low-abundance populations down to 0.01%. Based on a technically meticulous investigation, the authors make a convincing case for the ability of ligands to shift the WC/Hoogsteen equilibrium by DNA binding. This may eventually be of biological relevance, since there is evidence that Hoogsteen base pairing is involved in DNA target site binding by transcription factors such as p53, but the study falls short of making a convincing link to relevant biological problems. In its present form, the manuscript seems well suited for a Chemistry journal but may lack the broad appeal expected in an interdisciplinary journal.

Major comments:

1. MD simulations were not carried out for C·G base pairs, because their Hoogsteen form is associated with cytosine protonation, which is difficult to model in MD (page 11). Why is it not possible to start these simulations with C⁺·G base pairs? Would that not be equivalent to a conformational bias towards the Hoogsteen form? Please comment.

Minor comments:

2. On page 4, the lack of crystallographic evidence for Hoogsteen base-pair formation is explained by stating that "electron density associated with DNA bases is often weak". This is not correct, because in crystal structures the central bases pairs are nearly always better defined by electron density and have lower atomic displacement factors than other parts of a DNA structure. It would suffice to say that many structures are determined at a resolution that does not permit to detect low-abundance conformations.

3. On page 5, please replace "the principals of DNA" by "the principles of DNA".

4. On page 10, please replace "requires that a TA step to be sandwiched" by "requires that a TA step be sandwiched".

5. On page 16, please replace "read out mechanisms" by "readout mechanisms".

6. The list of references where 18 of 44 are dated before the year 2000 gives the impression of a study that is not extremely timely.

7. It seems I am fighting a lost battle here, but I wish to maintain that "data" is the plural form of "datum" and hence it must read "data were collected ..." and not "data was collected ...". The authors appear to be uncertain themselves, because they use both forms in the first paragraph on page 24.

8. Figure 1: Where do the quoted pWC values come from?

9. Figures 2 and 5: Do the color scales for WC/Hoogsteen abundance range from 0 to 100%?

10. Figure 6 largely repeats parts from other figures. It should be removed.

Reviewer #2:

Remarks to the Author:

This is an extremely interesting and timely paper that describes the influence of ligand binding on the Hoogsteen-WC base pair dynamics in DNA. The work represents an extension of earlier pioneering work from the Al-Hashimi lab showing that these types of transitions occur commonly in DNA, are dependent on nucleotide sequence, and often involved transiently-formed HG species that are only detectable using sophisticated relaxation dispersion NMR experiments. In this paper, Al-Hashimi and co-workers show that two antibiotics with different chemical and DNA binding properties, echinomycin and actinomycin D, both induce significant changes in WC-HG populations and transition dynamics, both at the site of binding and at adjacent sites. Some aspects of binding differ, and it is particularly interesting that actinomycin D binding leads to a minor species with chemical shifts that appear to be incompatible with shifts predicted for standard DNA structures. The authors do a nice job of relating their findings for these small molecule ligands to likely influences of HG-WC equilibria on protein:DNA recognition. The paper is well written and should be of broad general interest to groups studying DNA structure and recognition.

Reviewer #3:

Remarks to the Author:

Al-Hashimi and coauthors have used nuclear magnetic resonance relaxation dispersion and molecular dynamics simulations to study how Watson-Crick / Hoogsteen dynamics are modulated upon recognition of duplex DNA by the bisintercalator echinomycin and monointercalator actinomycin D. In both cases, DNA recognition results in the quenching of Hoogsteen dynamics at base pairs involved in intermolecular base-specific hydrogen bonds. In the case of echinomycin, the Hoogsteen population increased 10-fold for base pairs flanking the chromophore most likely due to intermolecular stacking interactions, whereas actinomycin D minimally affected Hoogsteen dynamics at other sites. This is an interesting study. Before considering publishing, I have a few questions

1. In the MD part, the biased simulations were applied to sample the transitions between WC and Hoogsteen base pairings. When a biased energy term is added, the dynamics of the system will be changed. Are these biased dynamics (or pathways) reliably representing the true dynamics of the transitions?

2. The energies shown in Fig.4b were calculated through the relative interaction energy for the bp that includes the flipping base as well as the bps above and below the flipping base as mentioned in the Method part. Is the interaction with the ligand, echinomycin, included? How about the solvent effects?

3. Simulation study was focused only on echinomycin molecule. How about the other ligand, actinomycin D? does it have similar effects as echinomycin?

Reviewer #1:

Xu and colleagues use NMR methods and MD simulations to study the phenomenon of “Hoogsteen breathing” in DNA which is characterized by a dynamic equilibrium between standard Watson-Crick (WC) base pairs and non-standard Hoogsteen base pairs. Their study is based on extensive prior work by the Al-Hashimi laboratory and others (in part going back to the 1980s) on the DNA duplex binding of the intercalating drugs echinomycin, actinomycin D and triostin A. In this prior work it was observed that ligand binding is associated with a stabilization of WC C-G base pairs enclosed by the bis-intercalator echinomycin and the formation of adjacent Hoogsteen base pairs.

After the basic structural features of echinomycin and actinomycin A DNA binding are established, the emphasis of the present study is on characterizing the WC/Hoogsteen dynamics and its modulation by intercalator binding. WC/Hoogsteen populations are measured by NMR relaxation dispersion (RD), a method that can detect short-lived low-abundance populations down to 0.01%. Based on a technically meticulous investigation, the authors make a convincing case for the ability of ligands to shift the WC/Hoogsteen equilibrium by DNA binding.

We thank the reviewer for his/her positive comments.

This may eventually be of biological relevance, since there is evidence that Hoogsteen base pairing is involved in DNA target site binding by transcription factors such as p53, but the study falls short of making a convincing link to relevant biological problems. In its present form, the manuscript seems well suited for a Chemistry journal but may lack the broad appeal expected in an interdisciplinary journal.

We feel that this first observation of Hoogsteen breathing modulation on DNA recognition is an important milestone that should be of broad interest to chemists seeking to design DNA-binding molecules; structural biologists and biophysicists interested in DNA structure-function relationships; and theoretical / computational scientists interested in modeling DNA. This is especially the case given that Hoogsteen breathing is a fundamental dynamic mode that occurs ubiquitously in duplex DNA of all sequence types, and that based on the findings reported in our manuscript, modulation of Hoogsteen breathing on DNA recognition is very likely to be a general process. We have tried to better clarify the importance of this work as well as its biological impact in the final paragraph of the introduction on page 4:

“Changes in the conformational dynamics following recognition of proteins and ribonucleic acids have been shown to play essential roles determining thermodynamic binding affinity²⁴, kinetics mechanisms of binding^{25,26} as well as downstream biological activity^{27,28}. Changes in Hoogsteen dynamics on recognition of DNA by proteins or small molecules could potentially be widespread and play important roles determining the binding affinity and specificity, rates of complex formation, as well as downstream biological activity. Here, as a first step toward examining whether or not Hoogsteen breathing is modulated on DNA recognition, we used NMR in concert with molecular dynamics simulations to characterize changes in Watson-Crick/Hoogsteen breathing that follows recognition by two different small molecules; echinomycin and actinomycin D. Our results expose a new layer of structural plasticity at the interfaces of DNA complexes that may have general and important implications for DNA recognition.”

While the focus of this initial study was to rigorously establish the modulation of Hoogsteen breathing on DNA recognition, which presents unique challenges relative to characterizing Hoogsteen breathing in naked DNA, the findings reported in our manuscript do have important biological implications, which are noted in the discussion section. These include 1) Hoogsteen modulation could potentially explain potent inhibition of VEGF and FLT1 expression by echinomycin binding to a sequence element similar to that studied in our manuscript and 2) identification of stacking as a new interaction to explain formation of Hoogsteen bps in protein-DNA complexes, and also the likelihood for having Hoogsteen breathing modulation occur in other protein-DNA complexes by similar mechanisms. We feel that a deeper evaluation of the biological role of Hoogsteen breathing modulation on DNA recognition in these and other biological areas is an important future step that is beyond the scope of the present study.

In the revised manuscript, we have also clarified some of the challenges in characterizing Hoogsteen breathing in the relevant protein-DNA complexes, and emphasized that this is an important next step in the second to final paragraph in the discussion section.

“Our study focused on DNA-small molecule complexes, in part because of technical difficulties in applying NMR RD experiments to larger protein-DNA complexes, but also because the simplicity of the systems afforded us a unique opportunity to verify that the observed chemical exchange does indeed correspond to Hoogsteen breathing within the context of the complex. Future studies should build on advances in RD methods to study larger protein systems⁵⁵ as well as employ other methods such as infrared spectroscopy⁴¹. This will provide insights into the importance of Hoogsteen modulation on DNA recognition and function.

Major comments:

1. MD simulations were not carried out for C·G base pairs, because their Hoogsteen form is associated with cytosine protonation, which is difficult to model in MD (page 11). Why is it not possible to start these simulations with C⁺·G base pairs? Would that not be equivalent to a conformational bias towards the Hoogsteen form? Please comment.

Separate simulation runs on either Watson-Crick (not protonated) or Hoogsteen (protonated) forms are in principle feasible for G-C bps; however, to compute even the differences in stability between Watson-Crick and Hoogsteen only from these separate simulations without the transitions between them requires intensive free energy perturbation (or thermodynamic integration) calculations which are beyond the scope of our simulation focus herein. Moreover, the transitions between the Watson-Crick and Hoogsteen states, in which we are most interested, requires more sophisticated techniques (e.g., empirical valence bond models extended to multiple protonation states) that can model the protonation events. These can be quite complex and involve multiple transfer steps. Thus, it is not feasible to model transitions based on simulations of two end points. We note that biased MD simulations between Watson-Crick and Hoogsteen have been reported by two other groups (ref 42 and 43) and both of them focused exclusively on A-T and not G-C. Similarly, our work in two prior publications focused on modeling A-T not G-C Hoogsteen transitions (ref 2 and 4).

We have clarified these points on page 11:

“Simulations were not carried out for C4-G9 and G5-C8 bps, because formation of G-C⁺ Hoogsteen bps is coupled to cytosine protonation⁷ through a process that could involve multiple proton transfer events that are difficult to model computationally in complex environments. Rather, all biased MD simulations of the Watson-Crick to Hoogsteen transition have focused on A-T not G-C^{42,43}. Note that while separate simulations could be conducted for the individual Hoogsteen or Watson-Crick bps, these would not provide information regarding the ease of their inter-conversion, which is the key information we are interested in obtaining from these computational studies.”

Minor comments:

2. On page 4, the lack of crystallographic evidence for Hoogsteen base-pair formation is explained by stating that “electron density associated with DNA bases is often weak”. This is not correct, because in crystal structures the central bases pairs are nearly always better defined by electron density and have lower atomic displacement factors than other parts of a DNA structure. It would suffice to say that many structures are determined at a resolution that does not permit to detect low-abundance conformations.

We made the suggested change.

3. On page 5, please replace “the principals of DNA” by “the principles of DNA”.

Done

4. On page 10, please replace “requires that a TA step to be sandwiched” by “requires that a TA step be sandwiched”.

Done

5. On page 16, please replace “read out mechanisms” by “readout mechanisms”.

Done

6. The list of references where 18 of 44 are dated before the year 2000 gives the impression of a study that is not extremely timely.

We have included references to more contemporary studies (refs 31, 42, 43, 49, 55) as well as new references (refs 24, 25, 26, 27, 28) that deal with the timely issues of dynamics and their role in recognition and function.

7. It seems I am fighting a lost battle here, but I wish to maintain that “data” is the plural form of “datum” and hence it must read “data were collected ...” and not “data was collected ...”. The authors appear to be uncertain themselves, because they use both forms in the first paragraph on page 24.

We followed the suggestion and now only refer to data in the plural form.

8. Figure 1: Where do the quoted pWC values come from?

These were reported in one of our prior publications (Nikolova et al. JACS, 2013). We now include a citation to this paper in the Figure legend.

“The abundances of both G-C and A-T Hoogsteen at neutral pH were taken from ref⁷.”

9. Figures 2 and 5: Do the color scales for WC/Hoogsteen abundance range from 0 to 100%?

We thank the reviewer for pointing this out. We have clarified this in the legends of Figures 2 and 5.

10. Figure 6 largely repeats parts from other figures. It should be removed.

We agree with the reviewer and have removed Figure 6.

Reviewer #2:

This is an extremely interesting and timely paper that describes the influence of ligand binding on the Hoogsteen-WC base pair dynamics in DNA. The work represents an extension of earlier pioneering work from the Al-Hashimi lab showing that these types of transitions occur commonly in DNA, are dependent on nucleotide sequence, and often involved transiently-formed HG species that are only detectable using sophisticated relaxation dispersion NMR experiments. In this paper, Al-Hashimi and co-workers show that two antibiotics with different chemical and DNA binding properties, echinomycin and actinomycin D, both induce significant changes in WC-HG populations and transition dynamics, both at the site of binding and at adjacent sites. Some aspects of binding differ, and it is particularly interesting that actinomycin D binding leads to a minor species with chemical shifts that appear to be incompatible with shifts predicted for standard DNA structures. The authors do a nice job of relating their findings for these small molecule ligands to likely influences of HG-WC equilibria on protein:DNA recognition. The paper is well written and should be of broad general interest to groups studying DNA structure and recognition.

We thank the reviewer for her/his positive assessment of our manuscript.

Reviewer #3:

Al-Hashimi and coauthors have used nuclear magnetic resonance relaxation dispersion and molecular dynamics simulations to study how Watson-Crick / Hoogsteen dynamics are modulated upon recognition of duplex DNA by the bisintercalator echinomycin and monointercalator actinomycin D. In both cases, DNA recognition results in the quenching of Hoogsteen dynamics at base pairs involved in intermolecular base-specific hydrogen bonds. In the case of echinomycin, the Hoogsteen population increased 10-fold for base pairs flanking the chromophore most likely due to intermolecular stacking interactions, whereas actinomycin D minimally affected Hoogsteen dynamics at other sites. This is an interesting study. Before considering publishing, I have a few questions

1. In the MD part, the biased simulations were applied to sample the transitions between WC and Hoogsteen base pairings. When a biased energy term is added, the dynamics of the system will be changed. Are these biased dynamics (or pathways) reliably representing the true dynamics of the transitions?

The trajectories generated by the biased force are indeed sampling the “true” potential energy surface of the system, so they are possible trajectories, except that we may have exaggerated their probabilities (meaning that they occur in the simulation on much faster time scales than they would occur in the true system, although the pathways visit likely states that the un-biased system would visit). The bias is necessary since we wish to sample millisecond dynamics, which is otherwise impossible via unbiased, direct MD. The bias employed is however such that when the system does progress towards the end state, we do not bias at all, and only back-moves are penalized via a gentle half-harmonic potential. Therefore, while time does advance artificially fast, we can say that the paths generated from our biased MD are likely paths taken by the base in its transition. Note that we have used this method to study Hoogsteen transitions in two previous studies, which are cited in the manuscript (ref 2 and 4).

2. The energies shown in Fig.4b were calculated through the relative interaction energy for the bp that includes the flipping base as well as the bps above and below the flipping base as mentioned in the Method part. Is the interaction with the ligand, echinomycin, included? How about the solvent effects?

In the calculation of the interaction energy, each atom of the base is evaluated individually for both bonded and non-bonded terms. The non-bonded terms for the CHARMM force field are the electrostatic and van der Waals ones. All atoms (within a long-enough cutoff) around the atoms selected will be included in the evaluation of that energy term. So by this method the interaction with the echinomycin and the solvent effects are included in the reported energies. We clarified this in the methods section on page 28:

“In this calculation, each atom of the base was evaluated individually for both bonded and non-bonded terms for the CHARMM force field, which includes the interaction with the ligand and the solvent effect”.

3. Simulation study was focused only on echinomycin molecule. How about the other ligand, actinomycin D? does it have similar effects as echinomycin?

As described in our response to Reviewer 1 modeling G-C Hoogsteen bps is complicated by the protonation of the cytosine. Modeling the actinomycin D complex is also complicated by the lack of availability of a crystal structure for the precise DNA sequence context used in our study. We clarified these points in the revised manuscript on page 16:

“The observed impact of actinomycin D binding on Hoogsteen breathing could not be examined computationally because of the difficulties in modeling formation of G-C⁺ Hoogsteen and because crystal structures are not available for the specific DNA sequence used here.”

We also clarify the choice of DNA sequence on page 14,

*“We studied the 1:1 complex between the DNA sequence 5'-AGATATGCATATCT-3' and actinomycin D. This palindromic sequence contains one favorable binding site but is also long enough to study effects on Hoogsteen breathing in neighboring bps. NMR analysis of this complex shows that it forms the previously described structure of a similar sequence⁵⁰ with all bps being Watson-Crick and with one actinomycin D bound to the G7C8 step (**Fig 5b**, **Supplementary Fig. 5**, and **Supplementary Note 3**).”*